# Design, synthesis, and biological activity of novel halogenated sulfite compounds

Yingshuai Liu[1,2], Guozhu Sheng[2], Baohong Liu[2], Ruofei Yin[2], Yaoyao Du[1,2], Bin Li [1]*

**1** Plant Protection College, Shenyang Agricultural University, Shenyang, Liaoning, China, **2** Shandong Kangqiao Biotechnology Co., Ltd, Binzhou, Shandong, China

* 13700021632@163.com

## Abstract

The acaricide propargite has been widely used for over 50 years without significant resistance issues. Addressing to the propargite defects of poor crop safety, thirty-six novel halogenated propargite analogues were designed, synthesized, and characterized using $^1$H NMR, $^{13}$C NMR spectroscopy, and HRMS. All target compounds were screened for activity against adult *Tetranychus cinnabarinus* (spider mites) and *Myzus persicae* (aphids). Two compounds exhibiting higher insecticidal activity were further evaluated for crop safety on cowpea seedlings. Structural modifications, such as replacing the *tert*-butyl group on the propargite benzene ring with chlorine or trifluoromethoxy, and substituting the propargyl group with fluorinated alkyl groups (e.g., 2-fluoroethyl or 3,3,3-trifluoropropyl), significantly enhanced both acaricidal and aphicidal activity. Compound **5.16** demonstrated superior acaricidal activity ($LC_{50}$: 14.85 mg L$^{-1}$) on *Tetranychus cinnabarinus* and excellent crop safety on cowpea seedlings. Additionally, Compound **5.32** exhibited both acaricidal ($LC_{50}$: 14.32 mg L$^{-1}$) and aphicidal activity, which is unusual in this chemical class. The compounds **5.16** and **5.32** could be used as promising leads for the discovery of novel acaricides or insecticides.

## Introduction

Leaf mites, characterized by their small body size, short generation cycles, strong reproductive capacity, and high egg-laying rates, pose a significant threat to agricultural productivity. Among the most harmful species, *Tetranychus cinnabarinus* has been particularly damaging to crop yields [1]. In recent years, controlling mite infestations has become increasingly challenging due to the rapid development of resistance to conventional acaricides. Notable examples of such acaricides include bifenazate [2,3], cyenopyrafen [3], spirodiclofen [4], etoxazole [5], avermectin [6], pyridaben [7], cyflumetofen [7–11], and fenpropathrin [12,13].

In contrast, propargite, an ATP inhibitor introduced to the market in 1969, has not exhibited significant resistance issues to date. However, its application is limited by its

**Data availability statement:** All relevant data are within the paper and its Supporting Information files.

**Funding:** The author(s) received no specific funding for this work.

**Competing interests:** The authors have declared that no competing interests exist.

phytotoxic effects on certain crops, particularly when applied under high-temperature conditions [14].

Recently, a novel pesticide, flupentiofenox, has been developed. This compound features a unique trifluoroethyl phenylsulfoxide structure and has demonstrated potent efficacy in reducing ATP levels in two-spotted spider mites at practical doses [15]. Additionally, flupentiofenox has shown high efficacy in controlling mites and other sucking insects, making it a promising candidate for integrated pest management strategies [16].

Many pharmaceuticals utilize halogenated molecules to facilitate the formation of halogen bonds with biomolecules, a strategy that has been widely documented in the literature [17–19]. Inspired by the presence of multiple halogen substitutions on both the benzene ring and the alkyl chain of the flupentiofenox molecule, we hypothesized that introducing halogen atoms into the propargite molecule could discover new molecules with better biological efficacy, as illustrated in Fig 1.

Acaricidal sulfite compounds were first reported by Rubber Company in 1964 [20,21]. Following this, Uniroyal Company disclosed a series of structurally similar compounds exhibiting notable acaricidal activity [22]. The majority of these early studies focused primarily on the modifications of propargite molecule to the hexane ring moiety [20–22]. However, there is limited literature available concerning the design, synthesis, and biological activity of propargite analogues, highlighting a significant gap in this area of research [23].

This study focused on the design and synthesis of a series of novel halogenated sulfite compounds. The synthetic route was illustrated in Scheme 1. Biological activity evaluations were conducted on adult *T. cinnabarinus* (spider mite) and *Myzus persicae* (green peach aphid), while crop safety assessments were performed on cowpea seedlings. The results demonstrated that several of the newly synthesized

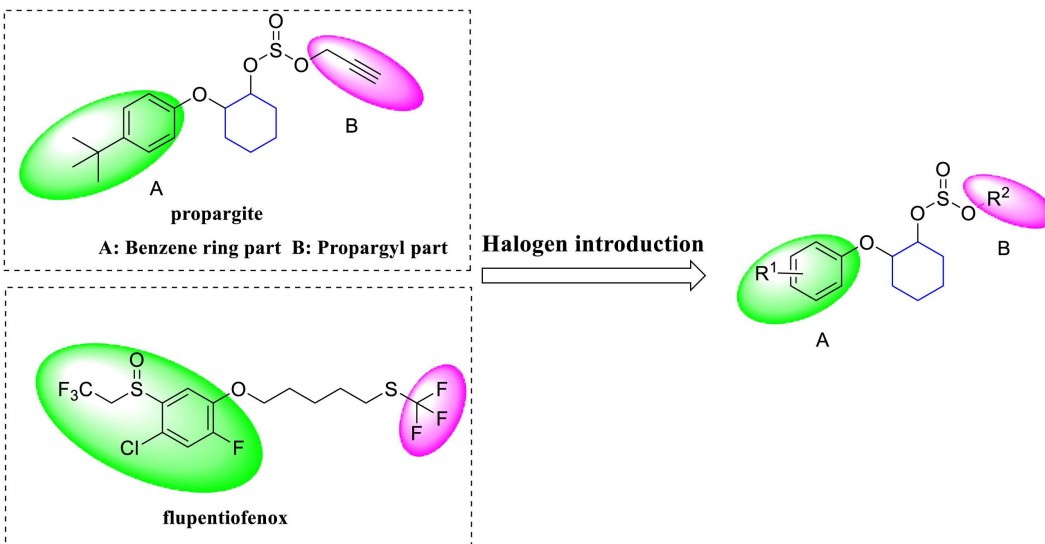

**Fig 1. Design of target compounds: introduction of halogens.**

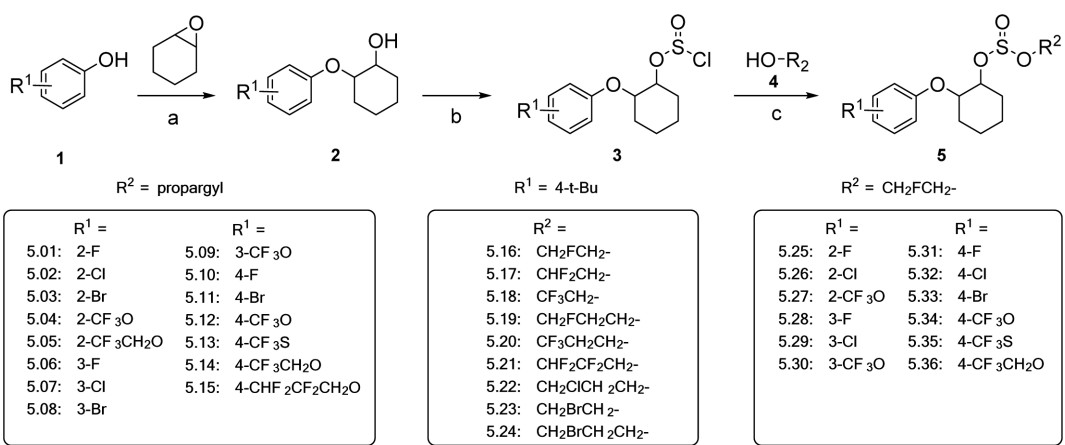

**Scheme 1. Synthetic route and target compounds 5.01-5.36 prepared.** Reagents and conditions: (a) NaOH, water, reflux; (b) toluene, 0 °C to r.t.; (c) Et₃N, toluene, 0 °C to r.t..

halogenated compounds exhibited significant acaricidal or aphicidal activity, while also proving to be safe for cowpea seedlings.

## Materials and methods

### Chemical synthesis

Reaction progress was monitored by thin-layer chromatography (TLC) on silica gel GF254 (Tianzhe, Qingdao, China), and spots were visualized with ultraviolet (UV) light (Gongyi, Zhengzhou, China). Column chromatography purification was performed using silica gel (200–300; Qingdao Haiyang Chemical, Qingdao, China). $^1$H NMR and $^{13}$C NMR spectra were recorded utilizing an JNM-ECZ600R spectrometer (JEOL, Tokyo, Japan), Bruker AVANCE 500 or 400 MHz spectrometer (Bruker Biospin, Rheinstetten, Germany) in CDCl$_3$ solution using tetramethylsilane as an internal reference standard at a temperature of 23–28°C. Chemical shifts were reported in ppm from the solvent resonance as the internal standard: CDCl$_3$ δ H = 7.26 ppm, δ C = 77.12 ppm; Multiplicity was indicated as follows: s (singlet), d (doublet), dd (doublet of doublets), t (triplet), q (quartet), ddd (doublet of double doublets), and m (multiplet); coupling constants were reported in Hz. High-resolution electrospray ionization mass spectra were obtained with an exactive focus mass spectrometer (LCQ Deca XP Max, Thermo Fisher, Waltham, USA; or 1290–6230 TOF, Agilent, CA, USA). High Performance Liquid Chromatography (HPLC) were performed with Agilent 1260 Infinity (Agilent Technologies, CA, USA).

All of the reagents and solvents, as well as propargite sample, are commercially available and were used directly without further purification.

**The preparation of intermediates 2.** All 2-(substituted-phenoxy)cyclohexan-1-ols **2** were synthesized according to the procedures described in the literature [24], and the reactions proceeded smoothly. The resulting intermediates **2** were used directly in subsequent reactions without the need of further purification.

**The general procedure for the synthesis of target compounds 5.01–5.36.** To a solution of 2-(substituted-phenoxy) cyclohexan-1-ols **2** (10.0 mmol) in toluene (25 mL), thionyl chloride (1.30 g, 11.0 mmol) was added dropwise at 0 °C with ice bath. The R$^2$-OH **4** (15.0 mmol) and triethylamine (1.50 g, 15.0 mmol) were then sequentially added dropwise over a period of 10 min after the reaction mixture had been stirred for 2 h. The resulting solution was further stirred at room temperature for 2 h until the reaction was complete, as monitored by TLC. The reaction solution was added into water (30 mL) and extracted with ethyl acetate (50 mL × 2). The combined organic layer was washed with brine, dried

over anhydrous sodium sulfate (2.0 g), and concentrated under vacuum. The residue was purified by silica gel column chromatography (fluent: EtOAc/hexane = 1/20, v/v) to afford the target compounds **5**.

The $^1$H NMR, $^{13}$C NMR Spectra, and HRMS of compounds **5.01**–**5.36** were provided in S1 File. The structure for each compound were provided in S3 File, and the primary NMR data files were provided in S4 File.

The yields and their structural characterization data of target compounds **5.01–5.36** were as follows.

*2-(2-fluorophenoxy)cyclohexyl prop-2-yn-1-yl sulfite* (**5.01**): colorless oil, yield, 62%; $^1$H NMR (400 MHz, Chloroform-*d*, c(concentration) = 60 mg/mL) δ 7.12–6.99 (m, 3H, Ph-H), 6.92 (ttd, *J* = 8.0, 4.7, 4.1, 2.2 Hz, 1H, Ph-H), 4.74–4.46 (m, 3H, OCH and OCH$_2$), 4.25–4.15 (m, 1H, OCH), 2.52 (dt, *J* = 13.6, 2.5 Hz, 1H, CCH), 2.24–2.08 (m, 2H, CH$_2$), 1.83–1.71 (m, 2H, CH$_2$), 1.71–1.45 (m, 2H, CH$_2$), 1.42–1.26 (m, 2H, CH$_2$); $^{13}$C NMR (126 MHz, Chloroform-*d*, c = 60 mg/mL) δ 123.33, 123.30, 121.49, 121.43, 121.32, 121.26, 117.46, 117.24, 115.73, 115.69, 115.58, 115.54, 79.49, 78.81, 75.11, 74.85, 74.67, 74.59, 47.23, 47.20, 30.23, 30.11, 28.70, 28.32, 22.36, 22.02, 21.88, 21.58. HRMS (ESI) m/z: Calcd for C$_{15}$H$_{17}$F-NaO$_4$S [M + Na]$^+$ 335.0724, found 335.0724.

*2-(2-chlorophenoxy)cyclohexyl prop-2-yn-1-yl sulfite* (**5.02**): yellow oil, yield, 63%; $^1$H NMR (400 MHz, Chloroform-*d*, c = 40 mg/mL) δ 7.28 (dd, *J* = 7.9, 1.6 Hz, 1H, Ph-H), 7.16–7.08 (m, 1H, Ph-H), 6.92 (td, *J* = 8.3, 7.8, 1.4 Hz, 1H, Ph-H), 6.83 (tt, *J* = 7.6, 1.6 Hz, 1H, Ph-H), 4.67–4.43 (m, 3H, OCH and OCH$_2$), 4.21 (dddd, *J* = 17.6, 9.8, 7.7, 4.2 Hz, 1H, OCH), 2.43 (dt, *J* = 12.1, 2.5 Hz, 1H, CCH), 2.23–1.96 (m, 2H, CH$_2$), 1.80–1.63 (m, 2H, CH$_2$), 1.63–1.42 (m, 2H, CH$_2$), 1.41–1.22 (m, 2H, CH$_2$); $^{13}$C NMR (101 MHz, Chloroform-*d*, c = 40 mg/mL) δ 153.05, 130.63, 130.61, 130.45, 127.69, 124.37, 122.31, 122.14, 116.11, 115.82, 79.54, 78.60, 75.76, 75.74, 75.19, 48.42, 48.12, 30.96, 30.77, 29.35, 28.81, 23.20, 22.73, 22.33. HRMS (ESI) m/z: Calcd for C$_{15}$H$_{17}$ClNaO$_4$S [M + Na]$^+$ 351.0428, found 351.0428.

*2-(2-bromophenoxy)cyclohexyl prop-2-yn-1-yl sulfite* (**5.03**): yellow oil, yield, 60%; $^1$H NMR (400 MHz, Chloroform-*d*, c = 40 mg/mL) δ 7.45 (dd, *J* = 7.9, 1.6 Hz, 1H, Ph-H), 7.23–7.13 (m, 1H, Ph-H), 6.90 (dd, *J* = 8.3, 1.4 Hz, 1H, Ph-H), 6.76 (td, *J* = 7.6, 1.4 Hz, 1H, Ph-H), 4.61–4.47 (m, 3H, OCH and OCH$_2$), 4.25 (ddd, *J* = 8.7, 7.2, 4.1 Hz, 1H, OCH), 2.42 (t, *J* = 2.5 Hz, 1H, CCH), 2.16–2.07 (m, 1H, CH$_2$), 2.07–1.95 (m, 1H, CH$_2$), 1.76–1.47 (m, 4H, CH$_2$), 1.42–1.23 (m, 2H, CH$_2$); $^{13}$C NMR (126 MHz, Chloroform-*d*, c = 40 mg/mL) δ 152.87, 152.83, 132.66, 132.63, 132.43, 127.39, 127.34, 121.66, 121.50, 114.67, 114.40, 112.64, 112.49, 78.22, 77.24, 76.57, 76.47, 74.74, 74.49, 73.80, 47.53, 47.13, 29.75, 29.64, 29.52, 28.64, 28.14, 27.57, 22.01, 21.88, 21.65, 21.57, 21.51, 21.13. HRMS (ESI) m/z: Calcd for C$_{15}$H$_{17}$BrNaO$_4$S [M + Na]$^+$ 394.9923, found 394.9926.

*prop-2-yn-1-yl (2-(2-(trifluoromethoxy)phenoxy)cyclohexyl) sulfite* (**5.04**): colorless oil, yield, 63%; $^1$H NMR (400 MHz, Chloroform-*d*, c = 20 mg/mL) δ 7.26–7.17 (m, 2H, Ph-H), 7.12–7.02 (m, 1H, Ph-H), 7.00–6.89 (m, 1H, Ph-H), 4.69–4.55 (m, 3H, OCH and OCH$_2$), 4.30 (ddd, *J* = 8.6, 7.1, 4.1 Hz, 1H, OCH), 2.53 (dt, *J* = 39.3, 2.5 Hz, 1H, CCH), 2.24–1.97 (m, 2H, CH$_2$), 1.83–1.70 (m, 2H, CH$_2$), 1.64–1.55 (m, 2H, CH$_2$), 1.50–1.30 (m, 2H, CH$_2$); $^{13}$C NMR (126 MHz, Chloroform-*d*, c = 20 mg/mL) δ 148.74, 138.19, 126.83, 122.26, 120.60, 120.46, 115.28, 115.16, 77.99, 77.11, 76.51, 74.72, 74.49, 47.49, 47.21, 29.85, 28.03, 27.52, 22.03, 21.55, 21.07. HRMS (ESI) m/z: Calcd for C$_{16}$H$_{17}$F$_3$NaO$_5$S [M + Na]$^+$ 401.0641, found 401.0601.

*prop-2-yn-1-yl (2-(2-(2,2,2-trifluoroethoxy)phenoxy)cyclohexyl) sulfite* (**5.05**): yellow oil, yield, 48%; $^1$H NMR (400 MHz, Chloroform-*d*, c = 40 mg/mL) δ 7.13–6.82 (m, 4H, Ph-H), 4.81–4.48 (m, 3H, OCH and OCH$_2$), 4.38 (qd, *J* = 8.4, 6.6 Hz, 2H, CF$_3$CH$_2$), 4.29–4.14 (m, 1H, OCH), 2.49 (dt, *J* = 13.2, 2.5 Hz, 1H, CCH), 2.29–2.07 (m, 2H, CH$_2$), 1.84–1.70 (m, 2H, CH$_2$), 1.70–1.30 (m, 4H, CH$_2$); $^{13}$C NMR (101 MHz, Chloroform-*d*, c = 40 mg/mL) δ 148.58, 148.55, 148.16, 148.04, 124.16, 124.05, 122.51, 122.35, 118.32, 117.89, 117.74, 117.57, 79.94, 79.24, 76.28, 75.72, 75.69, 75.56, 67.73, 48.48, 48.11, 31.13, 31.05, 29.59, 29.21, 23.34, 22.99, 22.84, 22.54. HRMS (ESI) m/z: Calcd for C$_{17}$H$_{19}$F$_3$NaO$_5$S [M + Na]$^+$ 415.0798, found 415.0797.

*2-(3-fluorophenoxy)cyclohexyl prop-2-yn-1-yl sulfite* (**5.06**): colorless oil, yield, 63%; $^1$H NMR (400 MHz, Chloroform-*d*, c = 60 mg/mL) δ 7.21 (td, *J* = 8.5, 6.8 Hz, 1H, Ph-H), 6.81–6.55 (m, 3H, Ph-H), 4.68–4.64 (m, 1H, OCH), 4.62–4.59 (m, 1H, OCH$_2$), 4.53–4.42 (m, 1H, OCH$_2$), 4.19 (ddt, *J* = 10.3, 8.1, 5.2 Hz, 1H, OCH), 2.70–2.46 (m, 1H, CCH), 2.29–2.06 (m,

2H, CH$_2$), 1.83–1.70 (m, 2H, CH$_2$), 1.70–1.53 (m, 1H, CH$_2$), 1.51–1.30 (m, 3H, CH$_2$); $^{13}$C NMR (101 MHz, Chloroform-*d*, c = 60 mg/mL) δ 164.86, 158.82, 158.71, 130.39, 130.29, 111.64, 111.61, 108.24, 108.03, 103.82, 103.58, 78.12, 76.30, 75.85, 75.70, 50.02, 48.31, 48.28, 31.17, 29.18, 23.12, 22.87, 22.64. HRMS (ESI) m/z: Calcd for C$_{15}$H$_{17}$FNaO$_4$S [M + Na]$^+$ 335.0724, found 335.0724.

*2-(3-chlorophenoxy)cyclohexyl prop-2-yn-1-yl sulfite* (**5.07**): colorless oil, yield, 55%; $^1$H NMR (400 MHz, Chloroform-*d*, c = 60 mg/mL) δ 7.27–7.07 (m, 1H, Ph-H), 7.03–6.86 (m, 2H, Ph-H), 6.86–6.71 (m, 1H, Ph-H), 4.76–4.45 (m, 3H, OCH and OCH$_2$), 4.29–4.08 (m, 1H, OCH), 2.62–2.48 (m, 1H, CCH), 2.22–2.04 (m, 2H, CH$_2$), 1.75 (tq, *J* = 7.0, 3.7 Hz, 2H, CH$_2$), 1.69–1.29 (m, 4H, CH$_2$); $^{13}$C NMR (126 MHz, Chloroform-*d*, c = 60 mg/mL) δ 157.16, 133.91, 129.37, 129.33, 120.64, 120.50, 115.58, 115.46, 113.54, 113.37, 77.61, 77.14, 74.97, 74.82, 74.64, 47.32, 47.26, 30.17, 30.13, 28.45, 28.14, 22.33, 22.08, 21.81, 21.60. HRMS (ESI) m/z: Calcd for C$_{15}$H$_{17}$ClNaO$_4$S [M + Na]$^+$ 351.0428, found 351.0433.

*2-(3-bromophenoxy)cyclohexyl prop-2-yn-1-yl sulfite* (**5.08**): yellow oil, yield, 59%; $^1$H NMR (400 MHz, Chloroform-*d*, c = 40 mg/mL) δ 7.10–6.97 (m, 3H, Ph-H), 6.84–6.75 (m, 1H, Ph-H), 4.63–4.46 (m, 2H, OCH$_2$), 4.46–4.35 (m, 1H, OCH), 4.17–4.06 (m, 1H, OCH), 2.46 (dt, *J* = 5.8, 2.5 Hz, 1H, CCH), 2.15–2.03 (m, 2H, CH$_2$), 1.78–1.63 (m, 2H, CH$_2$), 1.60–1.49 (m, 1H, CH$_2$), 1.43–1.22 (m, 3H, CH$_2$); $^{13}$C NMR (126 MHz, Chloroform-*d*, c = 40 mg/mL) δ 157.20, 129.70, 129.66, 123.58, 123.44, 118.49, 118.36, 114.05, 113.87, 77.63, 77.15, 75.97, 75.27, 74.97, 74.85, 74.63, 48.99, 47.35, 47.28, 30.17, 30.12, 28.44, 28.13, 22.32, 22.07, 21.81, 21.59. HRMS (ESI) m/z: Calcd for C$_{15}$H$_{17}$BrNaO$_4$S [M + Na]$^+$ 394.9923, found 394.9923.

*prop-2-yn-1-yl (2-(3-(trifluoromethoxy)phenoxy)cyclohexyl) sulfite* (**5.09**): colorless oil, yield, 59%; $^1$H NMR (400 MHz, Chloroform-*d*, c = 40 mg/mL) δ 7.26 (d, *J* = 8.3 Hz, 1H, Ph-H), 6.91–6.73 (m, 3H, Ph-H), 4.75–4.38 (m, 3H, OCH and OCH$_2$), 4.30–4.11 (m, 1H, OCH), 2.51 (dt, *J* = 8.3, 2.5 Hz, 1H, CCH), 2.16 (dddd, *J* = 15.0, 6.2, 4.5, 2.7 Hz, 2H, CH$_2$), 1.88–1.72 (m, 2H, CH$_2$), 1.67–1.33 (m, 4H, CH$_2$); $^{13}$C NMR (126 MHz, Chloroform-*d*, c = 40 mg/mL) δ 157.45, 149.14, 129.33, 129.28, 113.35, 113.18, 112.56, 112.44, 108.25, 108.14, 77.62, 77.13, 74.82, 74.80, 74.53, 47.37, 47.28, 30.11, 28.41, 28.10, 22.30, 22.04, 21.79, 21.56. HRMS (ESI) m/z: Calcd for C$_{16}$H$_{17}$F$_3$NaO$_5$S [M + Na]$^+$ 401.0641, found 401.0614.

*2-(4-fluorophenoxy)cyclohexyl prop-2-yn-1-yl sulfite* (**5.10**): colorless oil, yield, 63%; $^1$H NMR (400 MHz, Chloroform-*d*, c = 60 mg/mL) δ 7.04–6.79 (m, 4H, Ph-H), 4.77–4.54 (m, 2H, OCH$_2$), 4.47 (ddd, *J* = 10.3, 8.0, 4.6 Hz, 1H, OCH), 4.09 (dtd, *J* = 16.5, 8.9, 4.3 Hz, 1H, OCH), 2.51 (dt, *J* = 8.2, 2.5 Hz, 1H, CCH), 2.24–2.05 (m, 2H, CH$_2$), 1.77 (tt, *J* = 13.0, 6.2 Hz, 2H, CH$_2$), 1.67–1.56 (m, 1H, CH$_2$), 1.49–1.28 (m, 3H, CH$_2$); $^{13}$C NMR (101 MHz, Chloroform-d, c = 60 mg/mL) δ 158.84, 156.58, 156.46, 153.57, 117.97, 117.89, 117.66, 117.58, 116.11, 116.06, 115.88, 115.83, 79.79, 79.09, 77.49, 76.39, 75.95, 75.77, 48.27, 48.12, 31.18, 29.70, 29.33, 23.45, 23.17, 22.93, 22.68. HR-MS (ESI) m/z: Calcd for C$_{15}$H$_{17}$FNaO$_4$S [M + Na]$^+$ 335.0724, found 335.0724.

*2-(4-bromophenoxy)cyclohexyl prop-2-yn-1-yl sulfite* (**5.11**): colorless oil, yield, 62%; $^1$H NMR (400 MHz, Chloroform-*d*, c = 40 mg/mL) δ 7.41–7.29 (m, 2H, Ph-H), 6.87–6.78 (m, 2H, Ph-H), 4.74–4.53 (m, 2H, OCH$_2$), 4.48 (ddd, *J* = 10.3, 8.0, 4.5 Hz, 1H, OCH), 4.16 (qd, *J* = 8.4, 4.2 Hz, 1H, OCH), 2.61–2.48 (m, 1H, CCH), 2.15 (tt, *J* = 11.4, 4.5 Hz, 2H, CH$_2$), 1.76 (ddp, *J* = 13.0, 9.1, 4.4 Hz, 2H, CH$_2$), 1.68–1.57 (m, 1H, CH$_2$), 1.51–1.27 (m, 3H, CH$_2$); $^{13}$C NMR (101 MHz, Chloroform-d, c = 40 mg/mL) δ 156.59, 132.47, 132.43, 118.14, 117.94, 113.53, 78.87, 78.26, 76.14, 75.84, 75.68, 48.37, 48.23, 31.19, 31.12, 29.54, 29.19, 23.38, 23.10, 22.87, 22.62. HR-MS (ESI) m/z: Calcd for C$_{15}$H$_{17}$BrNaO$_4$S [M + Na]$^+$ 394.9923, found 394.9923.

*prop-2-yn-1-yl (2-(4-(trifluoromethoxy)phenoxy)cyclohexyl) sulfite* (**5.12**): colorless oil, yield, 63%; $^1$H NMR (400 MHz, Chloroform-*d*, c = 40 mg/mL) δ 7.13 (d, *J* = 8.6 Hz, 2H, Ph-H), 6.97–6.83 (m, 2H, Ph-H), 4.72–4.53 (m, 2H, OCH$_2$), 4.50 (ddd, *J* = 10.1, 7.8, 4.3 Hz, 1H, OCH), 4.17 (qd, *J* = 9.0, 4.4 Hz, 1H, OCH), 2.51 (dt, *J* = 9.3, 2.6 Hz, 1H, CCH), 2.16 (dt, *J* = 13.6, 6.6 Hz, 2H, CH$_2$), 1.84–1.70 (m, 2H, CH$_2$), 1.69–1.57 (m, 1H, CH$_2$), 1.40 (dddd, *J* = 38.5, 27.6, 13.7, 4.9 Hz, 3H, CH$_2$); $^{13}$C NMR (101 MHz, Chloroform-d, c = 40 mg/mL) δ 155.98, 143.14, 122.56, 122.52, 117.19, 116.95, 79.17, 78.52, 76.18, 75.82, 75.66, 48.39, 48.20, 31.20, 31.13, 29.58, 29.22, 23.39, 23.09, 22.88, 22.62. HR-MS (ESI) m/z: Calcd for C$_{16}$H$_{17}$F$_3$NaO$_5$S [M + Na]$^+$ 401.0641, found 401.0641.

*prop-2-yn-1-yl (2-(4-((trifluoromethyl)thio)phenoxy)cyclohexyl) sulfite* (**5.13**): colorless oil, yield, 56%; $^1$H NMR (400 MHz, Chloroform-d, c = 40 mg/mL) δ 7.56 (d, J = 8.3 Hz, 2H, Ph-H), 7.00–6.91 (m, 2H, Ph-H), 4.68–4.46 (m, 3H, OCH and OCH$_2$), 4.27 (td, J = 8.6, 4.2 Hz, 1H, OCH), 2.50 (t, J = 2.4 Hz, 1H, CCH), 2.23–2.10 (m, 2H, CH$_2$), 1.79 (dtd, J = 10.3, 7.2, 3.7 Hz, 2H, CH$_2$), 1.67 (dtd, J = 13.6, 10.0, 3.2 Hz, 1H, CH$_2$), 1.54–1.45 (m, 1H, CH$_2$), 1.44–1.32 (m, 2H, CH$_2$); $^{13}$C NMR (101 MHz, Chloroform-d, c = 40 mg/mL) δ 159.79, 138.38, 131.14, 128.07, 116.68, 115.40, 77.82, 75.88, 75.40, 48.50, 31.10, 29.11, 23.03, 22.57. HR-MS (ESI) m/z: Calcd for $C_{16}H_{17}F_3NaO_4S_2$ [M + Na]$^+$ 417.0413, found 417.0414.

*prop-2-yn-1-yl (2-(4-(2,2,2-trifluoroethoxy)phenoxy)cyclohexyl) sulfite* (**5.14**): colorless oil, yield, 48%; $^1$H NMR (400 MHz, Chloroform-d, c = 40 mg/mL) δ 6.94–6.83 (m, 4H, Ph-H), 4.70–4.53 (m, 2H, CF$_3$CH$_2$O), 4.47 (ddd, J = 10.1, 8.0, 4.5 Hz, 1H, OCH), 4.29 (q, J = 8.2 Hz, 2H, OCH$_2$), 4.09 (dddd, J = 12.6, 9.9, 8.0, 4.3 Hz, 1H, OCH), 2.51 (dt, J = 8.1, 2.5 Hz, 1H, CCH), 2.23–2.08 (m, 2H, CH$_2$), 1.83–1.70 (m, 2H, CH$_2$), 1.62–1.53 (m, 1H, CH$_2$), 1.50–1.25 (m, 3H, CH$_2$); $^{13}$C NMR (101 MHz, Chloroform-d, c = 40 mg/mL) δ 152.96, 152.36, 124.79, 122.03, 117.99, 117.67, 116.41, 116.38, 79.75, 79.02, 77.58, 76.43, 75.99, 75.75, 75.72, 67.02, 66.67, 48.25, 48.11, 31.21, 31.16, 29.75, 29.36, 23.45, 23.16, 22.92, 22.67. HR-MS (ESI) m/z: Calcd for $C_{17}H_{19}F_3NaO_5S$ [M + Na]$^+$ 415.0803, found 415.0804.

*prop-2-yn-1-yl (2-(4-(2,2,3,3-tetrafluoropropoxy)phenoxy)cyclohexyl) sulfite* (**5.15**): colorless oil, yield, 62%; $^1$H NMR (400 MHz, Chloroform-d, c = 20 mg/mL) δ 6.97–6.79 (m, 4H, Ph-H), 6.06 (tt, J = 53.1, 5.0 Hz, 1H, CHF$_2$CF$_2$), 4.72–4.53 (m, 2H, -CF$_2$CH$_2$O), 4.53–4.40 (m, 1H, OCH), 4.29 (ddt, J = 12.6, 11.5, 1.6 Hz, 2H, OCH$_2$), 4.16–3.99 (m, 1H, OCH), 2.51 (dt, J = 7.7, 2.5 Hz, 1H, CCH), 2.23–2.08 (m, 2H, CH$_2$), 1.84–1.70 (m, 2H, CH$_2$), 1.70–1.58 (m, 1H, CH$_2$), 1.49–1.26 (m, 3H, CH$_2$); $^{13}$C NMR (101 MHz, Chloroform-d, c = 20 mg/mL) δ 152.89, 152.83, 152.28, 152.14, 118.03, 117.72, 116.09, 116.06, 79.79, 79.05, 75.98, 75.75, 75.72, 66.53, 66.24, 65.94, 48.25, 48.10, 31.21, 31.15, 29.75, 29.36, 23.45, 23.16, 22.93, 22.67. HR-MS (ESI) m/z: Calcd for $C_{18}H_{20}F_4NaO_5S$ [M + Na]$^+$ 447.0865, found 447.0871.

*2-(4-(tert-butyl)phenoxy)cyclohexyl (2-fluoroethyl) sulfite* (**5.16**): colorless oil, yield, 65%; $^1$H NMR (600 MHz, Chloroform-d, c = 60 mg/mL) δ 7.31–7.27 (m, 2H, Ph-H), 6.90–6.81 (m, 2H, Ph-H), 4.65–4.37 (m, 3H, -CH$_2$F, OCH), 4.33–4.22 (m, 1H, OCH), 4.21–4.09 (m, 2H, OCH$_2$), 2.22–2.08 (m, 2H, CH$_2$), 1.82–1.69 (m, 2H, CH$_2$), 1.67–1.31 (m, 4H, CH$_2$CH$_2$), 1.28 (d, J = 1.1 Hz, 9H, C(CH$_3$)$_3$); $^{13}$C NMR (151 MHz, Chloroform-d, c = 60 mg/mL) δ 155.34, 155.22, 144.32, 144.20, 126.44 (d, J = 3.9 Hz), 115.99, 115.76, 82.36 (d, J = 8.8 Hz), 81.22 (d, J = 9.0 Hz), 78.81, 78.36, 77.35, 77.14, 76.92, 76.28, 59.33 (dd, J = 30.2, 20.8 Hz), 34.20, 31.59, 31.43 (d, J = 10.4 Hz), 29.89, 29.63, 23.57, 23.40, 22.97 (d, J = 19.6 Hz); $^{19}$F NMR (565 MHz, Chloroform-d) δ −223.68 – −224.14 (m). HR-MS (ESI) m/z: Calcd for $C_{18}H_{27}FNaO_4S$ [M + Na]$^+$ 381.1506, found 381.1506.

*2-(4-(tert-butyl)phenoxy)cyclohexyl (2,2-difluoroethyl) sulfite* (**5.17**): colorless oil, yield, 65%; $^1$H NMR (400 MHz, Chloroform-d, c = 40 mg/mL) δ 7.26–7.19 (m, 2H, Ph-H), 6.83–6.72 (m, 2H, Ph-H), 5.81 (tdt, J = 55.2, 20.9, 4.2 Hz, 1H, CHF$_2$), 4.38 (dddd, J = 25.0, 10.5, 8.2, 4.6 Hz, 1H, OCH), 4.25–4.04 (m, 2H, OCH$_2$), 3.97 (tdd, J = 12.4, 9.4, 4.7 Hz, 1H, OCH), 2.09 (dddd, J = 31.2, 13.8, 8.0, 4.7 Hz, 2H, CH$_2$), 1.77–1.64 (m, 2H, CH$_2$), 1.64–1.26 (m, 4H, CH$_2$CH$_2$), 1.22 (s, 9H, C(CH$_3$)$_3$); $^{13}$C NMR (101 MHz, Chloroform-d, c = 40 mg/mL) δ 154.09, 153.96, 143.38, 143.26, 125.39, 125.36, 114.87, 114.59, 112.19, 77.71, 77.39, 57.33, 57.02, 33.10, 30.47, 30.37, 28.85, 28.64, 22.54, 22.44, 21.96, 21.89. HR-MS (ESI) m/z: Calcd for $C_{18}H_{26}F_2NaO_4S$ [M + Na]$^+$ 399.1412, found 399.1413.

*2-(4-(tert-butyl)phenoxy)cyclohexyl (2,2,2-trifluoroethyl) sulfite* (**5.18**): colorless oil, yield, 68%; $^1$H NMR (400 MHz, Chloroform-d, c = 60 mg/mL) δ 7.30–7.19 (m, 2H, Ph-H), 6.88–6.70 (m, 2H, Ph-H), 4.42 (dddd, J = 23.5, 10.9, 8.3, 4.8 Hz, 1H, OCH), 4.34–4.20 (m, 1H, OCH), 4.18–3.98 (m, 2H, CH$_2$CF$_3$), 2.10 (dtd, J = 26.8, 8.7, 8.2, 3.9 Hz, 2H, CH$_2$), 1.70 (q, J = 10.1, 8.8 Hz, 2H, CH$_2$), 1.63–1.26 (m, 4H, CH$_2$CH$_2$), 1.22 (d, J = 1.7 Hz, 9H, C(CH$_3$)$_3$); $^{13}$C NMR (101 MHz, Chloroform-d, c = 60 mg/mL) δ 155.02, 154.91, 144.47, 144.32, 126.45, 126.41, 115.87, 115.56, 78.70, 78.24, 56.17, 34.13, 31.50, 31.45, 29.91, 29.64, 23.62, 23.45, 23.02, 22.89. HR-MS (ESI) m/z: Calcd for $C_{18}H_{25}F_3NaO_4S$ [M + Na]$^+$ 417.1318, found 417.1316.

*2-(4-(tert-butyl)phenoxy)cyclohexyl (3-fluoropropyl) sulfite* (**5.19**): yellow oil, yield, 49%; $^1$H NMR (400 MHz, Chloroform-d, c = 60 mg/mL) δ 7.28 (d, J = 8.4 Hz, 2H, Ph-H), 6.86 (d, J = 8.3 Hz, 2H, Ph-H), 4.69–4.34 (m, 3H, OCH and OCH$_2$), 4.34–4.11 (m, 2H, CH$_2$F), 4.03 (ddd, J = 24.3, 11.2, 6.3 Hz, 1H, OCH), 2.24–2.08 (m, 2H, CH$_2$), 2.00 (tp, J = 18.3,

6.1 Hz, 2H, CH$_2$), 1.76 (dq, $J$ = 10.8, 5.5, 5.0 Hz, 2H, CH$_2$), 1.68–1.32 (m, 4H, CH$_2$CH$_2$), 1.29 (s, 9H, C(CH$_3$)$_3$); $^{13}$C NMR (101 MHz, Chloroform-d, c = 60 mg/mL) δ 155.34, 155.27, 144.18, 144.04, 126.35, 126.32, 115.88, 115.61, 81.05, 79.40, 78.49, 78.39, 75.82, 75.61, 56.91, 56.86, 34.11, 31.52, 31.34, 30.62, 30.42, 29.71, 29.56, 23.37, 23.32, 22.84. HR-MS (ESI) m/z: Calcd for C$_{19}$H$_{29}$FNaO$_4$S [M + Na]$^+$395.1663, found 395.1662.

*2-(4-(tert-butyl)phenoxy)cyclohexyl (3,3,3-trifluoropropyl) sulfite* (**5.20**): colorless oil, yield, 63%; $^1$H NMR (400 MHz, Chloroform-d, c = 60 mg/mL) δ 7.31–7.26 (m, 2H, Ph-H), 6.86 (d, $J$ = 2.0 Hz, 1H, Ph-H), 6.84 (s, 1H, Ph-H), 4.51–4.22 (m, 2H, OCH), 4.22–4.00 (m, 2H, OCH$_2$), 2.43 (dtdd, $J$ = 23.4, 12.8, 10.1, 6.5 Hz, 2H, CH$_2$CF$_3$), 2.26–2.07 (m, 2H, CH$_2$), 1.84–1.71 (m, 2H, CH$_2$), 1.70–1.33 (m, 4H, CH$_2$CH$_2$), 1.29 (s, 9H, C(CH$_3$)$_3$); $^{13}$C NMR (101 MHz, Chloroform-d, c = 60 mg/mL) δ 155.24, 155.13, 144.20, 126.38, 126.36, 115.90, 115.61, 78.72, 78.57, 76.37, 53.31, 53.11, 34.47, 34.24, 34.18, 34.11, 31.50, 31.40, 29.86, 29.74, 23.50, 23.48, 22.96. HR-MS (ESI) m/z: Calcd for C$_{19}$H$_{27}$F$_3$NaO$_4$S [M + Na]$^+$ 431.1474, found 431.1473.

*2-(4-(tert-butyl)phenoxy)cyclohexyl (2,2,3,3-tetrafluoropropyl) sulfite* (**5.21**): colorless oil, yield, 66%; $^1$H NMR (400 MHz, Chloroform-d, c = 60 mg/mL) δ 7.34–7.25 (m, 2H, Ph-H), 6.84 (d, $J$ = 8.3 Hz, 2H, Ph-H), 5.78 (dtt, $J$ = 75.5, 53.0, 4.5 Hz, CHF$_2$), 4.58–4.26 (m, 2H, OCH$_2$), 4.26–4.07 (m, 2H, OCH), 2.32–2.05 (m, 2H, CH$_2$), 1.89–1.68 (m, 2H, CH$_2$), 1.67–1.32 (m, 4H, CH$_2$CH$_2$), 1.29 (s, 9H, C(CH$_3$)$_3$); $^{13}$C NMR (101 MHz, Chloroform-d, c = 60 mg/mL) δ 155.04, 154.88, 144.46, 144.31, 126.44, 126.41, 115.83, 115.43, 78.66, 78.37, 56.00, 55.70, 55.47, 34.13, 31.49, 31.38, 29.95, 29.66, 23.62, 23.51, 23.02, 22.94. HR-MS (ESI) m/z: Calcd for C$_{19}$H$_{26}$F$_4$NaO$_4$S [M + Na]$^+$ 449.1380, found 449.1380.

*2-(4-(tert-butyl)phenoxy)cyclohexyl (3-chloropropyl) sulfite* (**5.22**): colorless oil, yield, 65%; $^1$H NMR (400 MHz, Chloroform-d, c = 60 mg/mL) δ 7.28 (d, $J$ = 8.3 Hz, 2H, Ph-H), 6.86 (dd, $J$ = 8.7, 1.6 Hz, 2H, Ph-H), 4.44 (dddd, $J$ = 18.5, 10.3, 8.0, 4.5 Hz, 1H, OCH), 4.34–4.12 (m, 2H, OCH$_2$), 4.12–3.99 (m, 1H, OCH), 3.57 (dt, $J$ = 30.5, 6.4 Hz, 2H, CH$_2$Cl), 2.27–1.98 (m, 4H, CH$_2$), 1.85–1.69 (m, 2H, CH$_2$), 1.69–1.33 (m, 4H, CH$_2$CH$_2$), 1.29 (s, 9H, C(CH$_3$)$_3$); $^{13}$C NMR (101 MHz, Chloroform-d, c = 60 mg/mL) δ 155.33, 155.26, 144.21, 144.06, 126.36, 126.33, 115.89, 115.60, 78.53, 78.41, 75.96, 75.69, 57.70, 57.65, 40.95, 40.90, 34.12, 32.39, 31.52, 31.40, 29.74, 29.59, 23.37, 22.86. HR-MS (ESI) m/z: Calcd for C$_{19}$H$_{29}$ClNaO$_4$S [M + Na]$^+$ 411.1367, found 411.1367.

*2-bromoethyl (2-(4-tert-butyl)phenoxy)cyclohexyl) sulfite* (**5.23**): colorless oil, yield, 54%; $^1$H NMR (400 MHz, Chloroform-d, c = 60 mg/mL) δ 7.29 (d, $J$ = 8.4 Hz, 2H, Ph-H), 6.86 (d, $J$ = 8.3 Hz, 2H, Ph-H), 4.54–4.27 (m, 2H, OCH), 4.16 (ddt, $J$ = 17.6, 11.8, 6.7 Hz, 2H, OCH$_2$), 3.49 (t, $J$ = 6.4 Hz, 1H, CH$_2$Br), 3.43 (t, $J$ = 6.7 Hz, 1H, CH$_2$Br), 2.25–2.13 (m, 2H, CH$_2$), 1.84–1.71 (m, 2H, CH$_2$), 1.68–1.33 (m, 4H, CH$_2$CH$_2$), 1.29 (s, 9H, C(CH$_3$)$_3$); $^{13}$C NMR (101 MHz, Chloroform-d, c = 60 mg/mL) δ 155.25, 155.16, 144.15, 126.39, 126.37, 115.89, 115.63, 78.63, 78.41, 76.24, 60.15, 59.89, 34.12, 31.52, 31.48, 29.82, 29.66, 29.25, 29.00, 23.48, 23.40, 22.94, 22.89. HR-MS (ESI) m/z: Calcd for C$_{18}$H$_{27}$BrNaO$_4$S [M + Na]$^+$ 441.0706, found 441.0705.

*3-bromopropyl (2-(4-tert-butyl)phenoxy)cyclohexyl) sulfite* (**5.24**): colorless oil, yield, 56%; $^1$H NMR (400 MHz, Chloroform-d, c = 60 mg/mL) δ 7.28 (d, $J$ = 8.3 Hz, 2H, Ph-H), 6.93–6.79 (m, 2H, Ph-H), 4.44 (dddd, $J$ = 21.8, 10.0, 8.0, 4.5 Hz, 1H, OCH), 4.35–4.12 (m, 2H, OCH$_2$), 4.11–3.97 (m, 1H, OCH), 3.42 (dt, $J$ = 31.3, 6.2 Hz, 2H, CH$_2$Br), 2.15 (dp, $J$ = 18.7, 6.4 Hz, 4H, CH$_2$CH$_2$), 1.85–1.68 (m, 2H, CH$_2$), 1.69–1.32 (m, 4H, CH$_2$CH$_2$), 1.29 (s, 9H, C(CH$_3$)$_3$); $^{13}$C NMR (101 MHz, Chloroform-d, c = 60 mg/mL) δ 155.26, 126.37, 126.34, 115.90, 115.60, 78.54, 78.43, 76.01, 75.71, 58.70, 58.65, 34.12, 32.49, 31.53, 31.43, 29.76, 29.60, 29.25, 29.18, 23.37, 22.87. HR-MS (ESI) m/z: Calcd for C$_{19}$H$_{29}$BrNaO$_4$S [M + Na]$^+$ 455.0862, found 455.0861.

*2-fluoroethyl (2-(2-fluorophenoxy)cyclohexyl) sulfite* (**5.25**): colorless oil, yield, 65%; $^1$H NMR (500 MHz, Chloroform-d, c = 60 mg/mL) δ 7.06–6.93 (m, 3H, Ph-H), 6.89–6.81 (m, 1H, Ph-H), 4.62–4.51 (m, 1H, OCH), 4.51–4.36 (m, 2H, OCH$_2$), 4.31–4.07 (m, 3H, OCH and CH$_2$F), 2.17–2.05 (m, 2H, CH$_2$), 1.78–1.67 (m, 2H, CH$_2$), 1.64–1.40 (m, 2H, CH$_2$), 1.37–1.19 (m, 2H, CH$_2$); $^{13}$C NMR (126 MHz, Chloroform-d, c = 60 mg/mL) δ 153.68, 151.72, 144.32, 123.36, 123.33, 121.40, 121.35, 121.23, 121.17, 117.32, 116.96, 115.70, 115.66, 115.54, 115.51, 81.37, 81.30, 80.00, 79.93, 79.49, 78.92, 75.09, 74.90, 58.63, 58.46, 58.33, 58.16, 30.42, 30.39, 28.78, 28.52, 22.41, 22.20, 21.93, 21.77. HRMS (ESI) m/z: Calcd for C$_{14}$H$_{18}$F$_2$NaO$_4$S [M + Na]$^+$ 343.0786, found 343.0796.

*2-(2-chlorophenoxy)cyclohexyl (2-fluoroethyl) sulfite* (**5.26**): yellow oil, yield, 65%; [1]H NMR (400 MHz, Chloroform-*d*, c = 60 mg/mL) δ 7.26 (dd, *J* = 7.9, 1.7 Hz, 1H, Ph-H), 7.11 (ddt, *J* = 8.8, 7.4, 1.6 Hz, 1H, Ph-H), 6.92 (ddd, *J* = 8.4, 5.4, 1.4 Hz, 1H, Ph-H), 6.82 (tt, *J* = 7.6, 1.4 Hz, 1H, Ph-H), 4.67–4.32 (m, 3H, OCH and OCH$_2$), 4.32–4.00 (m, 3H, OCH and CH$_2$F), 2.18–1.97 (m, 2H, CH$_2$), 1.79–1.62 (m, 2H, CH$_2$), 1.63–1.40 (m, 2H, CH$_2$), 1.40–1.22 (m, 2H, CH$_2$); [13]C NMR (126 MHz, Chloroform-*d*, c = 60 mg/mL) δ 152.68, 129.50, 129.41, 126.71, 126.54, 123.44, 121.29, 121.02, 115.86, 84.52, 83.81, 83.75, 83.20, 80.84, 80.74, 74.19, 72.34, 61.18, 61.03, 31.09, 30.92, 28.73, 28.60, 23.71, 23.02, 22.99, 22.90, 22.79, 22.62. HRMS (ESI) m/z: Calcd for C$_{14}$H$_{18}$ClFNaO$_4$S [M + Na]$^+$ 359.0491, found 359.0497.

*2-fluoroethyl (2-(2-(trifluoromethoxy)phenoxy)cyclohexyl) sulfite* (**5.27**): yellow oil, yield, 64%; [1]H NMR (400 MHz, Chloroform-*d*, c = 40 mg/mL) δ 7.21–7.11 (m, 2H, Ph-H), 6.99 (td, *J* = 8.9, 8.2, 1.5 Hz, 1H, Ph-H), 6.88 (ddt, *J* = 9.0, 7.7, 1.4 Hz, 1H, Ph-H), 4.67–4.36 (m, 3H, OCH and OCH$_2$), 4.33–4.00 (m, 3H, OCH and CH$_2$F), 2.19–1.98 (m, 2H, CH$_2$), 1.76–1.63 (m, 2H, CH$_2$), 1.60–1.40 (m, 2H, CH$_2$), 1.37–1.18 (m, 2H, CH$_2$); [13]C NMR (126 MHz, Chloroform-*d*, c = 40 mg/mL) δ 148.79, 138.04, 126.87, 122.26, 122.23, 120.55, 120.44, 115.26, 115.11, 81.31, 81.21, 79.95, 79.85, 78.11, 77.46, 74.36, 74.19, 58.59, 58.54, 58.43, 58.37, 30.05, 29.95, 28.17, 27.82, 22.12, 21.76, 21.65, 21.36. HRMS (ESI) m/z: Calcd for C$_{15}$H$_{18}$F$_4$NaO$_5$S [M + Na]$^+$ 409.0703, found 409.0695.

*2-fluoroethyl (2-(3-fluorophenoxy)cyclohexyl) sulfite* (**5.28**): colorless oil, yield, 60%; [1]H NMR (400 MHz, Chloroform-*d*, c = 40 mg/mL) δ 7.21 (td, *J* = 8.2, 6.8 Hz, 1H, Ph-H), 6.81–6.55 (m, 3H, Ph-H), 4.72–4.59 (m, 1H, OCH), 4.56–4.37 (m, 2H, OCH$_2$), 4.34–4.04 (m, 3H, OCH and CH$_2$F), 2.17 (dddd, *J* = 19.0, 13.0, 5.6, 3.1 Hz, 2H, CH$_2$), 1.77 (tdt, *J* = 8.9, 5.6, 2.8 Hz, 2H, CH$_2$), 1.69–1.29 (m, 4H, CH$_2$CH$_2$); [13]C NMR (126 MHz, Chloroform-*d*, c = 40 mg/mL) δ 163.58, 161.63, 129.38, 129.35, 129.30, 129.28, 110.81, 110.78, 110.65, 107.30, 107.20, 107.13, 107.03, 102.91, 102.77, 102.71, 102.58, 81.32, 81.25, 79.95, 79.89, 77.72, 77.38, 74.89, 74.76, 58.46, 58.42, 58.30, 58.25, 30.32, 30.23, 28.57, 28.32, 22.36, 22.18, 21.86, 21.72. HRMS (ESI) m/z: Calcd for C$_{14}$H$_{18}$F$_2$NaO$_4$S [M + Na]$^+$ 343.0786, found 343.0786.

*2-(3-chlorophenoxy)cyclohexyl (2-fluoroethyl) sulfite* (**5.29**): colorless oil, yield, 50%; [1]H NMR (500 MHz, Chloroform-*d*, c = 40 mg/mL) δ 7.12 (t, *J* = 8.1 Hz, 1H, Ph-H), 6.92–6.81 (m, 2H, Ph-H), 6.78–6.70 (m, 1H, Ph-H), 4.62–4.49 (m, 1H, OCH), 4.49–4.30 (m, 2H, OCH$_2$), 4.28–3.98 (m, 3H, OCH and CH$_2$F), 2.16–2.01 (m, 2H, CH$_2$), 1.79–1.64 (m, 2H, CH$_2$), 1.57 (dddd, *J* = 13.7, 11.5, 10.1, 3.8 Hz, 1H, CH$_2$), 1.47–1.20 (m, 3H, CH$_2$); [13]C NMR (126 MHz, Chloroform-*d*, c = 40 mg/mL) δ 157.22, 133.91, 129.36, 129.34, 120.62, 120.52, 115.46, 113.58, 113.44, 81.25, 79.88, 77.78, 77.46, 74.76, 58.44, 58.27, 30.32, 30.23, 28.68, 28.57, 28.33, 22.36, 22.19, 21.87, 21.74. HRMS (ESI) m/z: Calcd for C$_{14}$H$_{18}$ClFNaO$_4$S [M + Na]$^+$ 359.0491, found 359.0490.

*2-fluoroethyl (2-(3-(trifluoromethoxy)phenoxy)cyclohexyl) sulfite* (**5.30**): colorless oil, yield, 55%; [1]H NMR (400 MHz, Chloroform-*d*, c = 20 mg/mL) δ 7.26 (d, *J* = 8.3 Hz, 1H, Ph-H), 6.91–6.72 (m, 3H, Ph-H), 4.71–4.39 (m, 3H, OCH and OCH$_2$), 4.34–4.06 (m, 3H, OCH and CH$_2$F), 2.23–2.05 (m, 2H, CH$_2$), 1.78 (tdd, *J* = 7.9, 4.1, 2.1 Hz, 2H, CH$_2$), 1.69–1.56 (m, 1H, CH$_2$), 1.48–1.29 (m, 3H, CH$_2$); [13]C NMR (126 MHz, Chloroform-*d*, c = 20 mg/mL) δ 157.51, 129.29, 113.20, 112.46, 108.15, 81.20, 79.84, 77.41, 58.48, 58.31, 30.29, 28.28, 22.33, 22.14, 21.69. HRMS (ESI) m/z: Calcd for C$_{15}$H$_{18}$F$_4$NaO$_5$S [M + Na]$^+$ 409.0703, found 409.0703.

*2-fluoroethyl (2-(4-fluorophenoxy)cyclohexyl) sulfite* (**5.31**): colorless oil, yield, 66%; [1]H NMR (400 MHz, Chloroform-*d*, c = 40 mg/mL) δ 6.96 (t, *J* = 8.6 Hz, 2H, Ph-H), 6.88 (dd, *J* = 9.1, 4.3 Hz, 2H, Ph-H), 4.69–4.56 (m, 1H, OCH), 4.55–4.48 (m, 1H, OCH$_2$), 4.43 (ddd, *J* = 10.2, 7.9, 4.7 Hz, 1H, OCH$_2$), 4.34–3.99 (m, 3H, OCH and CH$_2$F), 2.14 (dh, *J* = 12.5, 4.0 Hz, 2H, CH$_2$), 1.85–1.69 (m, 2H, CH$_2$), 1.61 (ddd, *J* = 13.8, 10.5, 3.9 Hz, 1H, CH$_2$), 1.50–1.25 (m, 3H, CH$_2$); [13]C NMR (101 MHz, Chloroform-d, c = 40 mg/mL) δ 158.86, 156.48, 153.62, 117.74, 117.66, 116.06, 115.84, 82.48, 80.77, 79.39, 76.03, 59.41, 59.20, 31.36, 29.49, 23.26, 22.79. HR-MS (ESI) m/z: Calcd for C$_{14}$H$_{18}$F$_2$NaO$_4$S [M + Na]$^+$ 343.0786, found 343.0786.

*2-(4-chlorophenoxy)cyclohexyl (2-fluoroethyl) sulfite* (**5.32**): yellow oil, yield, 66%; [1]H NMR (400 MHz, Chloroform-*d*, c = 40 mg/mL) δ 7.28–7.19 (m, 2H, Ph-H), 6.91–6.83 (m, 2H, Ph-H), 4.68–4.58 (m, 1H, OCH), 4.57–4.39 (m, 2H, OCH$_2$), 4.35–4.01 (m, 3H, OCH and CH$_2$F), 2.15 (dq, *J* = 13.7, 4.5 Hz, 2H, CH$_2$), 1.76 (dtt, *J* = 13.2, 9.7, 4.5 Hz, 2H, CH$_2$), 1.61 (ddt, *J* = 17.4, 11.3, 3.2 Hz, 1H, CH$_2$), 1.52–1.26 (m, 3H, CH$_2$); [13]C NMR (101 MHz, Chloroform-d, c = 40 mg/mL) δ 156.23,

156.13, 129.50, 129.48, 126.40, 126.26, 117.73, 117.53, 82.51, 82.45, 80.75, 79.15, 78.67, 76.07, 75.82, 59.49, 59.45, 59.28, 59.25, 31.31, 31.24, 29.66, 29.37, 23.40, 23.20, 22.91, 22.74. HR-MS (ESI) m/z: Calcd for $C_{14}H_{18}ClFNaO_4S$ [M＋Na]⁺ 359.0491, found 359.0491.

*2-(4-bromophenoxy)cyclohexyl (2-fluoroethyl) sulfite* (**5.33**): colorless oil, yield, 63%; ¹H NMR (400 MHz, Chloroform-*d*, c＝40 mg/mL) δ 7.44–7.29 (m, 2H, Ph-H), 6.81 (dd, *J*＝9.0, 2.4 Hz, 2H, Ph-H), 4.69–4.57 (m, 1H, OCH), 4.56–4.37 (m, 2H, OCH₂), 4.34–3.97 (m, 3H, OCH and CH₂F), 2.14 (dh, *J*＝13.5, 5.3, 4.3 Hz, 2H, CH₂), 1.76 (dhept, *J*＝12.8, 4.0 Hz, 2H, CH₂), 1.62 (dddd, *J*＝17.1, 11.3, 8.7, 3.4 Hz, 1H, CH₂), 1.49–1.25 (m, 3H, CH₂); ¹³C NMR (101 MHz, Chloroform-d, c＝40 mg/mL) δ 156.74, 156.65, 132.46, 132.43, 118.19, 117.99, 113.71, 113.55, 82.51, 82.45, 80.80, 80.74, 79.03, 78.55, 76.03, 75.78, 59.50, 59.47, 59.29, 31.30, 31.23, 29.63, 29.35, 23.39, 23.19, 22.90, 22.73. HR-MS (ESI) m/z: Calcd for $C_{14}H_{18}BrFNaO_4S$ [M＋Na]⁺ 402.9985, found 402.9985.

*2-fluoroethyl (2-(4-(trifluoromethoxy)phenoxy)cyclohexyl) sulfite* (**5.34**): colorless oil, yield, 55%; ¹H NMR (400 MHz, Chloroform-*d*, c＝20 mg/mL) δ 7.10–7.01 (m, 2H, Ph-H), 6.90–6.81 (m, 2H, Ph-H), 4.61–4.52 (m, 1H, OCH), 4.52–4.34 (m, 2H, OCH₂), 4.28–4.14 (m, 1H, OCH), 4.11 (ddd, *J*＝7.1, 5.3, 3.5 Hz, 1H, CH₂F), 4.08–3.98 (m, 1H, CH₂F), 2.14–2.02 (m, 2H, CH₂), 1.77–1.65 (m, 2H, CH₂), 1.61–1.53 (m, 1H, CH₂), 1.44–1.22 (m, 3H, CH₂); ¹³C NMR (101 MHz, Chloroform-d, c＝20 mg/mL) δ 156.04, 122.51, 119.29, 117.02, 82.43, 80.72, 78.80, 75.75, 59.52, 59.32, 31.30, 29.37, 23.18, 22.73. HR-MS (ESI) m/z: Calcd for $C_{15}H_{18}F_4NaO_5S$ [M＋Na]⁺ 409.0703, found 409.0713.

*2-fluoroethyl (2-(4-((trifluoromethyl)thio)phenoxy)cyclohexyl) sulfite* (**5.35**): colorless oil, yield, 60%; ¹H NMR (400 MHz, Chloroform-*d*, c＝40 mg/mL) δ 7.59–7.52 (m, 2H, Ph-H), 6.95 (dd, *J*＝9.0, 3.2 Hz, 2H, Ph-H), 4.62 (dq, *J*＝16.0, 3.5, 2.6 Hz, 1H, OCH), 4.58–4.43 (m, 2H, OCH₂), 4.34–4.02 (m, 3H, OCH and CH₂F), 2.17 (ddq, *J*＝13.7, 9.2, 4.7 Hz, 2H, CH₂), 1.85–1.71 (m, 2H, CH₂), 1.70–1.58 (m, 1H, CH₂), 1.53–1.30 (m, 3H, CH₂); ¹³C NMR (101 MHz, Chloroform-d, c＝40 mg/mL) δ 159.85, 138.39, 116.82, 116.70, 115.42, 82.47, 82.39, 80.68, 78.47, 78.07, 75.89, 75.56, 59.62, 59.51, 59.41, 59.30, 31.29, 31.23, 29.56, 29.29, 23.35, 23.14, 22.87, 22.70. HR-MS (ESI) m/z: Calcd for $C_{15}H_{18}F_4NaO_4S_2$ [M＋Na]⁺ 425.0475, found 425.0475.

*2-fluoroethyl (2-(4-(2,2,2-trifluoroethoxy)phenoxy)cyclohexyl) sulfite* (**5.36**): colorless oil, yield, 55%; ¹H NMR (400 MHz, Chloroform-*d*, c＝20 mg/mL) δ 6.98–6.78 (m, 4H, Ph-H), 4.70–4.67 (m, 1H, OCH₂CF₃), 4.66–4.59 (m, 1H, OCH₂CF₃), 4.58–4.54 (m, 1H, OCH), 4.50–4.42 (m, 1H, OCH), 4.31–4.28 (m, 2H, OCH₂), 4.24–4.20 (m, 1H, CH₂F), 4.09 (dddd, *J*＝18.1, 8.2, 5.6, 3.1 Hz, 1H, CH₂F), 2.14 (ddt, *J*＝14.1, 9.6, 4.1 Hz, 2H, CH₂), 1.76 (qt, *J*＝8.2, 6.0, 4.7 Hz, 2H, CH₂), 1.63 (ddt, *J*＝19.3, 6.0, 3.2 Hz, 1H, CH₂), 1.48–1.26 (m, 3H, CH₂); ¹³C NMR (101 MHz, Chloroform-*d*, c＝20 mg/mL) δ 152.96, 152.26, 124.79, 118.03, 117.75, 116.41, 82.52, 82.48, 82.32, 80.61, 79.88, 79.32, 76.25, 76.06, 67.04, 66.69, 61.22, 61.01, 59.40, 59.19, 31.33, 31.25, 29.81, 29.52, 23.44, 23.24, 22.93, 22.78. HR-MS (ESI) m/z: Calcd for $C_{16}H_{20}F_4NaO_5S$ [M＋Na]⁺ 423.0865, found 423.0866.

## Biological activity and safety testing on cowpea seedlings

**Acaricidal activity evaluation against *T. cinnabarinus*.** Biological assays against adults of *T. cinnabarinus* were conducted according to procedures described in the literature [25–27]. All prepared compounds as well as the propargite were dissolved in acetone and diluted with water containing 0.1% Tween 80 to give solutions with different concentrations. One true leaf of kidney bean plants was infested with 110–130 adult *T. cinnabarinus* individuals. The test solutions (1 mL) were sprayed using an airbrush at 25±1°C. Once it was dry, the leaf blade was transferred to a standard observation room in a green house. The commercial acaricide propargite was used as a positive control at the same time. A blank control was treated as above without the addition of compounds. The mortality of the mite under the action of compounds was evaluated after 72 h of exposure. Each treatment was repeated three times. Evaluation of mortality was based on a percentage scale of 0–100, in which 0 equals no activity, and 100 equals total lethality. Mortality rates were corrected using Abbott's formula [28]. Median lethal concentration (LC₅₀) values were calculated by probit analysis [29].

$$\text{Corrected mortality rate } (\%) = \frac{T - C}{1 - C} \times 100$$

T was the mortality rate of the tested compound group, and C was the mortality rate of the blank control group (both T and C were expressed as percentages). The statistical analysis of all acaricidal activity assays was calculated using IBM SPSS Statistics 23.0.

**Aphicidal activity evaluation against _Myzus persicae_.** Biological assays against _M. persicae_ were conducted according to procedures described in the literature [30,31]. A single true leaf (3–5 cm in length) of rapeseed plants was selected and infested with 25–35 second- to third-instar nymphs of _M. persicae_ (green peach aphid). At 25 ± 1°C, the test solutions (2 mL) (prepared as described in 3.3.1) were sprayed onto both sides of the leaf using a trigger-type spray bottle. The treated leaf was then placed in a sealed petri dish and transferred to a standard observation room in a greenhouse. The commercial acaricide propargite was used as a positive control at the same time. Two blank controls—a solvent control (acetone + 0.1% Tween 80 solution) and an environmental control (untreated)—were processed following the same protocol without the addition of test compounds. After 72 hours of incubation at 25 ± 1°C, aphid mortality was assessed. Each treatment was repeated three times. Evaluation of mortality was based on a percentage scale of 0–100, in which 0 equals no activity, and 100 equals total lethality. Mortality rates were corrected using Abbott's formula [28].

**Preliminary safety evaluation on cowpea seedlings.** The test solutions (1 mL) (prepared as described in 3.3.1) were sprayed using an airbrush on the cowpea seedlings with newly expanded leaves. The commercial acaricide propargite was used as a positive control at the same time. A blank control was treated as above without the addition of compounds. Treated cowpea seedlings were placed in a constant temperature incubator at 35 °C. After 24 h, the cowpea seedlings were transferred to a standard observation room in a green house and maintained growing for 48 h. The damage to the plants was evaluated visually [32,33].

## Results and discussion

### The synthesis of Compounds 5.01–5.36

The synthesis of the target compounds (**5.01–5.36**), as outlined in Scheme 1, was straightforward and proceeded smoothly. The yield for each step in the synthetic route typically exceeded 70%. Following purification via column chromatography, the purity of all target compounds, as determined by HPLC and $^1$H NMR analysis, ranged between 95% and 99%. The structures of the target compounds were characterized using $^1$H NMR, $^{13}$C NMR spectroscopy and HRMS.

### The acaricidal activity against adult _T. cinnabarinus_

As presented in Table 1, most of the target compounds demonstrated acaricidal activities against adult _T. cinnabarinus_ at 200 mg/L.

The median lethal concentration (LC$_{50}$) values for the highest seven target compounds, together with propargite, were provided in Table 2.

On the benzene ring (R$^1$ part), replacing of the _tert_-butyl group with chlorine or trifluoromethoxy preserved the acaricidal activity. For the same group such as trifluoromethoxy, the 4-position substitution showed the highest activity, followed by 3-position substitution and 2-position substitution. Introduction of fluorinated alkyl groups in place of the propargyl group (R$^2$ part) resulted in enhanced acaricidal activity.

Based on the acaricidal activity results in Tables 1 and 2, the acaricidal structure-activity relationship of the target compounds were summarized in Fig 2.

Five novel compounds showed higher acaricidal activity than that of propargite (LC$_{50}$ = 26.94 mg/L). Keeping the same benzene ring part as propargite, the compound **5.16** (LC$_{50}$ = 14.85 mg/L) was chosen for further crop safety study.

**The aphicidal activity against _M. persicae_.** Considering the broad insecticidal spectrum of flupentiofenox, all the target compounds were screened for aphicidal activity. As presented in Table 3, twelve compounds demonstrated better aphicidal activities against _M. persicae_ than that of propargite at 100 mg/L.

**Table 1. Mortalities of the target compounds against adult *Tetranychus cinnabarinus*.**

| Compound | Mortality rate (mean±SD, %) (mg L$^{-1}$) | | | | |
|---|---|---|---|---|---|
| | 200 | 100 | 50 | 25 | 12.5 |
| **Propargite** | 99.5±0.7 | 95.6±1.8 | 76.1±5.4 | 48.0±4.1 | 15.2±7.6 |
| **5.01** | 97.7±0.7 | 77.0±5.3 | 45.0±9.3 | —[a] | — |
| **5.02** | 99.4±0.8 | 97.2±0.5 | 76.0±3.6 | — | — |
| **5.03** | 95.2±0.4 | 80.4±1.5 | 33.5±1.2 | — | — |
| **5.04** | 100±0.0 | 99.2±1.1 | 68.2±4.6 | — | — |
| **5.05** | 100±0.0 | 92.2±1.3 | 73.6±5.8 | — | — |
| **5.06** | 96.4±1.4 | 45.9±7.5 | 41.4±2.2 | — | — |
| **5.07** | 99.5±0.7 | 98.3±0.4 | 67.0±5.5 | — | — |
| **5.08** | 97.8±0.4 | 82.1±1.1 | 78.4±6.3 | — | — |
| **5.09** | 100±0.0 | 95.8±1.3 | 87.3±3.2 | 42.0±11.4 | 23.4±4.4 |
| **5.10** | 97.1±1.0 | 88.8±4.5 | 66.7±8.9 | — | — |
| **5.11** | 94.7±1.5 | 90.2±3.0 | 73.9±4.1 | — | — |
| **5.12** | 100±0.0 | 97.3±2.7 | 83.9±2.8 | 67.4±4.1 | 29.6±9.8 |
| **5.13** | 41.7±8.2 | 10.3±3.4 | 0±0 | — | — |
| **5.14** | 100±0.0 | 90.2±0.6 | 59.2±0.7 | — | — |
| **5.15** | 52.6±4.4 | 38.5±5.9 | 15.7±4.2 | — | — |
| **5.16** | 100±0.0 | 100±0.0 | 90.2±2.0 | 67.1±4.4 | 41.9±6.4 |
| **5.17** | 98.8±0.8 | 93.1±1.3 | 66.3±2.6 | — | — |
| **5.18** | 94.8±0.6 | 89.9±0.8 | 55.2±3.3 | — | — |
| **5.19** | 77.8±4.2 | 43.7±11.9 | 11.1±4.3 | — | — |
| **5.20** | 100±0.0 | 97.6±0.8 | 93.6±0.8 | 71.0±6.6 | 43.7±0.6 |
| **5.21** | 86.0±6.7 | 55.3±6.6 | 24.0±18.1 | — | — |
| **5.22** | 80.9±3.6 | 67.1±1.8 | 36.8±7.5 | — | — |
| **5.23** | 88.1±3.1 | 76.1±3.3 | 47.8±5.7 | — | — |
| **5.24** | 65.1±3.0 | 29.6±1.0 | 6.4±4.4 | — | — |
| **5.25** | 96.7±1.1 | 92.2±1.7 | 59.8±10.0 | — | — |
| **5.26** | 100±0.0 | 95.5±2.3 | 68.1±4.4 | — | — |
| **5.27** | 100±0.0 | 98.8±0.8 | 79.8±3.5 | 50.8±1.2 | 20.8±3.2 |
| **5.28** | 99.5±0.7 | 96.5±0.8 | 72.7±3.9 | — | — |
| **5.29** | 98.1±0.1 | 85.4±6.2 | 66.0±6.2 | — | — |
| **5.30** | 100±0.0 | 92.3±4.4 | 82.6±7.6 | — | — |
| **5.31** | 100±0.0 | 94.6±1.4 | 69.5±0.6 | — | — |
| **5.32** | 98.1±0.4 | 97.0±1.2 | 87.6±2.6 | 81.5±3.6 | 34.6±5.1 |
| **5.33** | 90.6±1.7 | 80.2±1.2 | 54.7±3.3 | — | — |
| **5.34** | 100±0.0 | 97.4±1.1 | 88.4±1.0 | 72.7±4.2 | 44.6±1.9 |
| **5.35** | 73.9±6.2 | 47.1±8.3 | 14.1±3.6 | — | — |
| **5.36** | 100±0.0 | 93.3±1.0 | 63.1±9.0 | — | — |

[a]"—" refers to "not tested".

Introduction fluorinated alkyl groups in place of the propargyl group (R$^2$ part) promoted aphicidal activity significantly. The most of 2-fluoroethyl derivatives showed good activity. On the benzene ring (R$^1$ part), replacement of the *tert*-butyl group with chlorine and fluorinated alkyl groups strengthened the aphicidal activity.

Based on the aphicidal activity results in Table 3, the aphicidal structure-activity relationship of the target compounds were summarized in Fig 3.

**Table 2. Median lethal concentration (LC$_{50}$) values for target compounds against adult _T. cinnabarinus_.**

| Compound | LC$_{50}$ (mg L$^{-1}$) | 95% confidence interval | Formula ($y=ax+b$) | Correlation coefficient |
|---|---|---|---|---|
| Propargite | 26.94 | 24.84 - 29.14 | $y=2.95x-4.23$ | 0.999 |
| 5.09 | 23.98 | 17.55 - 31.26 | $y=2.91x-4.01$ | 0.969 |
| 5.12 | 18.79 | 16.74 - 20.79 | $y=2.70x-3.45$ | 0.988 |
| 5.16 | 14.85 | 12.33 - 17.55 | $y=2.28x-2.68$ | 0.993 |
| 5.20 | 14.50 | 12.49 - 16.40 | $y=2.44x-2.81$ | 0.983 |
| 5.27 | 24.23 | 22.11 - 26.39 | $y=3.31x-4.55$ | 0.980 |
| 5.32 | 14.32 | 2.34 - 23.88 | $y=1.96x-2.19$ | 0.920 |
| 5.34 | 14.29 | 11.89 - 16.45 | $y=2.29x-2.63$ | 0.998 |

**Fig 2. The acaricidal SAR of synthesized halogenated propargite analogues.**

**Table 3. Mortalities of the target compounds against _M. persicae_.**

| Compound | Mortality rate (%) (mg L$^{-1}$) | | | Compound | Mortality rate (%) (mg L$^{-1}$) | | | |
|---|---|---|---|---|---|---|---|---|
| | 400 | 100 | 25 | | 400 | 100 | 25 | |
| Propargite | 77.1 | 0 | 0 | 5.19 | 0 | — | — | |
| 5.01 | 0 | —[a] | — | 5.20 | 0 | — | — | |
| 5.02 | 75.7 | — | — | 5.21 | 0 | — | — | |
| 5.03 | 0 | — | — | 5.22 | 0 | — | — | |
| 5.04 | 94.2 | 0 | 0 | 5.23 | 0 | — | — | |
| 5.05 | 0 | — | — | 5.24 | 0 | — | — | |
| 5.06 | 83.9 | — | — | 5.25 | 100 | 89.8 | 66.1 | |
| 5.07 | 0 | — | — | 5.26 | 100 | 89.3 | 0 | |
| 5.08 | 0 | — | — | 5.27 | 99.1 | 95.8 | 46.2 | |
| 5.09 | 96.3 | 0 | 0 | 5.28 | 100 | 100 | 83.9 | |
| 5.10 | 96.4 | 0 | 0 | 5.29 | 83.1 | 0 | 0 | |
| 5.11 | 71.3 | 0 | 0 | 5.30 | 100 | 84.6 | 0 | |
| 5.12 | 0 | — | — | 5.31 | 100 | 94.8 | 0 | |
| 5.13 | 100 | 64.8 | 26.4 | 5.32 | 100 | 100 | 57.3 | |
| 5.14 | 71 | — | — | 5.33 | 99.1 | 88.7 | 0 | |
| 5.15 | 79.9 | — | — | 5.34 | 79.1 | 0 | 0 | |
| 5.16 | 96.8 | 58.6 | 0 | 5.35 | 100 | 100 | 71.2 | |
| 5.17 | 0 | — | — | 5.36 | 100 | 100 | 95.7 | |
| 5.18 | 0 | — | — | | | | | |

[a]"—" refers to "not tested".

**Fig 3. The aphicidal SAR of synthesized halogenated propargite analogues.**

At 100 mg/L, the compound **5.32** showed both higher aphicidal activity (mortality rate: 100%, Table 3) and higher acaricidal activity (mortality rate: 97%, Table 1). Considering its cost-effective advantage, the compound **5.32** was chosen for further crop safety study.

## Safety on cowpea seedlings

The recommended field application dosage for the commercial acaricide propargite is approximately 500 mg/L. Over the past decade, occasional crop injuries have been reported, particularly when applied at high temperatures [34–37].

The compound **5.16** with higher acaricidal activity, and compound **5.32** with higher acaricidal activity and higher aphicidal activity, were carried out for safety evaluation on cowpea seedlings.

The test results, as is shown in Fig 4 and 5, indicated that compound **5.32** damaged cowpea seedlings slightly at 1000 mg/L. However, at a higher dosage of 2000 mg/L, significant damage to the cowpea seedlings was observed. Compared with CK (photo 2D), the leaf blades treated with compound **5.32** wrinkled seriously (photo 2C). The safety of compound **5.32** and propargite on cowpea seedlings were found to be equivalent.

It was an encouraging finding in that compound **5.16** is safe to cowpea seedlings (photos 1B and 2B). Even at a high concentration of 2000 mg/L, compound **5.16** did not cause significant damage to the cowpea seedlings compared with CK, indicated that 2-fluoroethyl may have a positive effect on decreasing phytotoxicity of this type of compound. The compound **5.16** exhibits superior biological efficacy compared to the commercial acaricide propargite because of its higher acaricidal activity and better crop safety.

The original photos were provided in S2 File.

## Conclusion

A series of sulfite compounds were designed by combining the molecular structures of propargite and flupentiofenox. Thirty-six novel halogenated sulfite compounds were synthesized and characterized using $^1$H NMR, $^{13}$C NMR spectroscopy, and HRMS.

Structure-activity relationship (SAR) studies of the halogenated sulfite compounds revealed that replacing the *tert*-butyl group on the benzene ring (R$^1$ part) of propargite with chlorine or trifluoromethoxy, as well as introducing 2-fluoroethyl groups in place of the propargyl group (R$^2$ part), significantly enhanced both acaricidal and aphicidal activity.

Among the synthesized compounds, compound **5.16** exhibited superior acaricidal activity and demonstrated good crop safety on cowpea seedlings. Although compound **5.32** did not show improved safety on cowpea seedlings, it displayed both acaricidal and aphicidal activity, which is unusual in this chemical class.

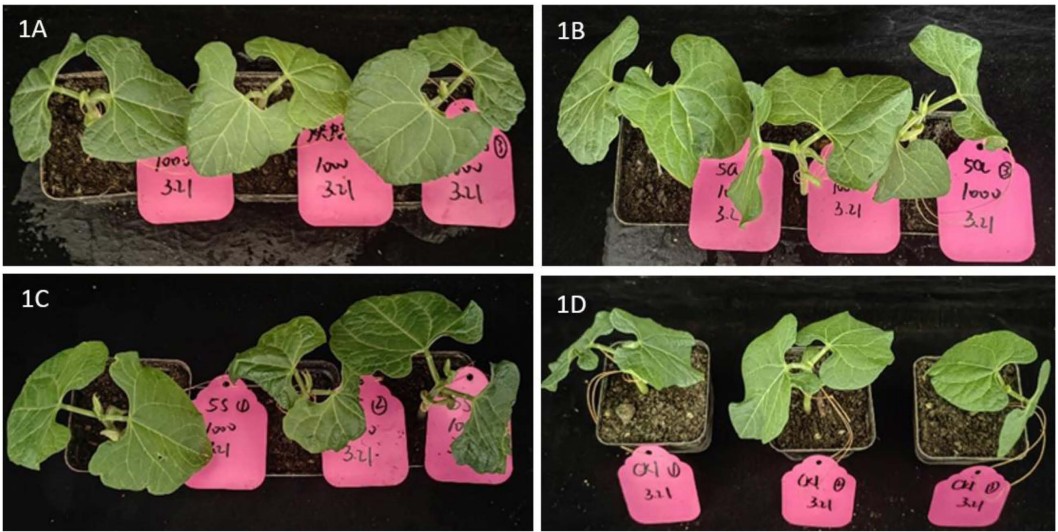

**Fig 4. Safety of compounds on cowpea seedlings at mass concentrations of 1000 mg/L.** 1A: propargite; 1B: compound **5.16**; 1C: compound **5.32**; 1D: CK.

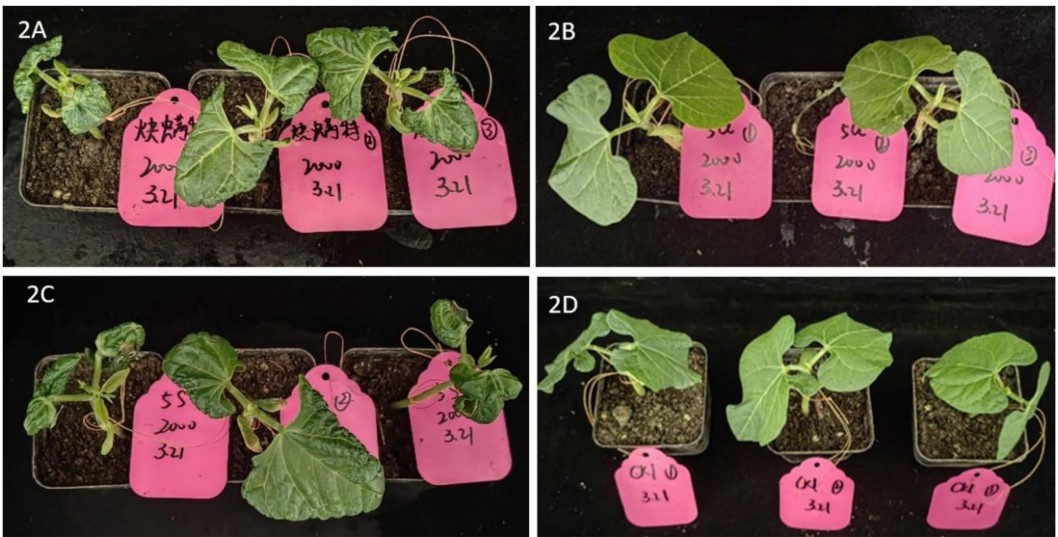

**Fig 5. Safety of compounds on cowpea seedlings at mass concentrations of 2000 mg/L.** 2A: propargite; 2B: compound **5.16**; 2C: compound **5.32**; 2D: CK.

In conclusion, compounds **5.16** and **5.32** could be used as promising leads for the discovery of novel acaricides or insecticides.

## Supporting information

**S1 File. The ¹H NMR, ¹³C NMR spectra, and HRMS of compounds 5.01–5.36.**
(DOCX)

**S2 File. The original photos of cowpea seedlings treated with compounds and CK.**
(DOCX)

**S3 File. Structure file for each compound.**
(DOCX)

**S4 File. The primary NMR data files.**
(ZIP)

## Author contributions

**Conceptualization:** Bin Li.

**Investigation:** Yingshuai Liu, Guozhu Sheng, Baohong Liu, Ruofei Yin, Yaoyao Du.

**Methodology:** Yingshuai Liu, Yaoyao Du.

**Supervision:** Bin Li.

**Validation:** Yingshuai Liu, Guozhu Sheng, Baohong Liu, Ruofei Yin, Yaoyao Du.

**Writing – original draft:** Yingshuai Liu.

**Writing – review & editing:** Bin Li.

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
