## [Decision Letter · Decision Letter 0]

Dear Dr. Li,

Thank you for submitting your manuscript to PLOS ONE. After careful consideration, we feel that it has merit but does not fully meet PLOS ONE’s publication criteria as it currently stands. Therefore, we invite you to submit a revised version of the manuscript that addresses the points raised during the review process.

We look forward to receiving your revised manuscript.

Kind regards,

Komal Rizwan, PhD

Academic Editor

PLOS ONE

2. We note that this submission includes NMR spectroscopy data. We would recommend that you include the following information in your methods section or as Supporting Information files:

1) The make/source of the NMR instrument used in your study, as well as the magnetic field strength. For each individual experiment, please also list: the nucleus being measured; the sample concentration; the solvent in which the sample is dissolved and if solvent signal suppression was used; the reference standard and the temperature.

2) A list of the chemical shifts for all compounds characterised by NMR spectroscopy, specifying, where relevant: the chemical shift (δ), the multiplicity and the coupling constants (in Hz), for the appropriate nuclei used for assignment.

3)The full integrated NMR spectrum, clearly labelled with the compound name and chemical structure.

We also strongly encourage authors to provide primary NMR data files, in particular for new compounds which have not been characterised in the existing literature. Authors should provide the acquisition data, FID files and processing parameters for each experiment, clearly labelled with the compound name and identifier, as well as a structure file for each provided dataset. See our list of recommended repositories here: https://journals.plos.org/plosone/s/recommended-repositories .

Additional Editor Comments (if provided):

Reviewers' comments:

Reviewer's Responses to Questions

**Comments to the Author**

1. Is the manuscript technically sound, and do the data support the conclusions?

Reviewer #1: Yes

Reviewer #2: Yes

2. Has the statistical analysis been performed appropriately and rigorously?

Reviewer #1: Yes

Reviewer #2: Yes

3. Have the authors made all data underlying the findings in their manuscript fully available?

Reviewer #1: Yes

Reviewer #2: Yes

4. Is the manuscript presented in an intelligible fashion and written in standard English?

Reviewer #1: Yes

Reviewer #2: Yes

Reviewer #1: Review Report

• The manuscript details the design, synthesis, and assessment of 36 halogenated sulfite derivatives, drawing inspiration from the structures of propargite and flupentiofenox—an innovative and pertinent strategy in the development of acaricides and aphicides.

• The research illustrates distinct structure–activity relationships (SARs), indicating how particular substitutions (e.g., Cl, CF₃O, 2-fluoroethyl) augment bioactivity, supported by robust chemical rationale and justification.

• Compounds 5.16 and 5.32 are prominently identified as lead candidates. Compound 5.32 exhibits remarkable and uncommon dual acaricidal and aphicidal properties within this class of compounds.

• The analytical characterization is sufficient and substantiates the structural assertions.

• The biological screening is well-structured; however, additional information regarding assay conditions (replicates, controls, statistical analysis) would enhance reproducibility.

• The assessment of crop safety for cowpea seedlings is pertinent; however, it is advisable to include additional discussion on broader phytotoxicity and environmental safety.

• The conclusion is robust but would improve with a concise discussion of the mechanism of action, prospective research, and potential applications.

• The language is generally clear; however, a brief editing session is advisable to enhance the flow and expression in the abstract and conclusion.

Minor Revisions — The manuscript is scientifically robust and offers significant insights; several improvements in clarity, depth, and contextual framing will enhance the final version.

Reviewer #2: Comments:

In this work the authors were designed, synthesized thirty-six halogenated propargite analogues and characterized using 1H NMR, 13C NMR spectroscopy, and HRMS. Among these thirty-six compounds two compounds exhibiting good insecticidal activity. Therefore, through SAR design authors discovered two interesting lead compounds (compounds 5.16 and 5.32) for insecticides.I recommend to publish in the PLOS ONE after minor revision.

Other comments:

1. I have not seen chemical structure at Figure 1. Design of target compounds: introduction of halogens

2. Scheme 1. Try to mention on the arrow “a” “b” and “c”

3. You write one number after digit in the 13C data

**Do you want your identity to be public for this peer review?** For information about this choice, including consent withdrawal, please see our Privacy Policy

Reviewer #1: No

Reviewer #2: No

---

## [Author Response · Author response to Decision Letter 1]

13 Jun 2025

Dear Komal Rizwan,

I have upload the revised manuscript and those related files. Please help me to make sure of that I am the last author.

Best Regards!

Bin

---

## [Editor Report · Decision Letter 1]

Design, synthesis, and biological activity of novel halogenated sulfite compounds

PONE-D-25-15070R1

Dear Author,

We’re pleased to inform you that your manuscript has been judged scientifically suitable for publication and will be formally accepted for publication once it meets all outstanding technical requirements.

Kind regards,

Komal Rizwan, PhD

Academic Editor

PLOS ONE
---

## [Editor Report · Acceptance letter]

PONE-D-25-15070R1

PLOS ONE

Dear Dr. Li,

I'm pleased to inform you that your manuscript has been deemed suitable for publication in PLOS ONE. Congratulations! Your manuscript is now being handed over to our production team.

Kind regards,

on behalf of

Dr. Komal Rizwan

Academic Editor

PLOS ONE